# Does Cardiorespiratory Fitness Moderate the Association between Occupational Stress, Cardiovascular Risk, and Mental Health in Police Officers?

**DOI:** 10.3390/ijerph16132349

**Published:** 2019-07-03

**Authors:** René Schilling, Flora Colledge, Sebastian Ludyga, Uwe Pühse, Serge Brand, Markus Gerber

**Affiliations:** 1Department of Sport, Exercise and Health, University of Basel, 4052 Basel, Switzerland; 2Center for Affective, Stress and Sleep Disorders, Psychiatric Clinics of the University of Basel, 4052 Basel, Switzerland

**Keywords:** cardiorespiratory fitness, cardiovascular health, psychosocial stress, police officers, mental health

## Abstract

*Background:* Chronic exposure to occupational stress may lead to negative health consequences. Creating less stressful work environments and making employees physically and psychologically more resilient against stress are therefore two major public health concerns. This study examined whether cardiorespiratory fitness moderated the association between occupational stress, cardiovascular risk, and mental health. *Methods:* Stress was assessed via the Effort-Reward Imbalance and Job Demand-Control models in 201 police officers (36% women, Mage = 38.6 years). Higher levels of blood pressure, blood lipids, blood sugar, and unfavorable body composition were considered as cardiovascular risk factors. Burnout, insomnia and overall psychological distress were used as mental health indicators. Cardiorespiratory fitness was assessed with a submaximal bicycle test. *Results:* High cardiorespiratory fitness levels were associated with a reduced cardiometabolic risk, whereas high stress levels were associated with better mental health. Among participants who perceived a high Effort-Reward Imbalance, those with high fitness levels showed lower overall cardiovascular risk scores than their colleagues with low fitness levels. *Conclusions:* Work health programs for police officers should consider the early screening of burnout, sleep disturbances, and overall mental wellbeing. To increase cardiovascular health, including fitness tests in routine health checks and promoting physical activity to further increase cardiorespiratory fitness appears worthwhile.

## 1. Introduction

Prolonged exposure to stressful life circumstances that exceed individuals’ coping capacities can result in emotional, cognitive, physiological and somatic health impairments [1]. While most individuals are able to handle short-term stress, chronic stress can have severe health consequences [2]. Studies show that high levels of psychosocial stress are a risk factor for cardiovascular diseases (CVDs), with an impact comparable to smoking or diabetes, and that a dose-response relationship exists between psychosocial stress and premature death [3,4].

Adverse work conditions constitute a major source of distress for many adults [5]. Despite multiple efforts during the past decade, work-related stress has not substantially decreased in European countries [6]. Accordingly, costs associated with occupational stress (e.g., due to health care, absenteeism, presenteeism, reduced productivity, increased turnover) remain high, and can be estimated at 0.5 to 1.2% of the gross domestic product (GDP) [7].

In the occupational stress literature, two of the most prominent theoretical models are the job demands-control (JDC) model [8] and the effort-reward imbalance (ERI) model [9]. Broadly speaking, the JDC model assumes that stress occurs when workers perceive a discrepancy between the demands they face at work and their ability to control those requirements with their existing resources. In the JDC model, such an imbalance is labelled as “job strain”. By contrast, the ERI model assumes that employees perceive their work as stressful, if they have the feeling that they invest a lot of energy (efforts), but do not get enough recognition (reward) for their efforts. Empirical evidence supports the validity of both models showing that an imbalance between demands/control and efforts/reward is associated with an increased risk for impaired mental wellbeing (e.g., burnout, anxiety, impaired sleep quality), decreased physical health (e.g., coronary heart disease, musculoskeletal disease), and premature death [10,11].

According to the World Health Organization [12], CVDs are the leading cause of deaths worldwide. In their review of prospective epidemiologic studies, Siegrist and Dragano [11] highlighted that high job strain doubles the risk for CVDs, particularly among men. Similarly, Nyberg et al. [13] found in a meta-analytic survey that work-related stress was linked to an increased risk for CVDs, mainly due to an elevated risk for type II diabetes, smoking, and physical inactivity. In line with this notion, the American Heart Association provided recommendations on the most relevant factors to maintain cardiovascular health [14]. These include: body mass index <25; ≥30 min of moderate intensity physical activity per day; <200 mg/dL of total blood cholesterol; ≤120/80 mmHg blood pressure; and ≤100 mg/dL fasting blood glucose [14]. As highlighted by Mottillo et al. [15], if individuals fail to meet several of these recommendations, their risk of all-cause mortality substantially increases.

Additionally, ample evidence exists for a close link between occupational stress and decreased mental wellbeing [16]. In high-income countries, the social burden of mental disorders is high, with costs estimated at 2.3 to 4.4% of the GDP [17]. Based on their review of prospective epidemiologic studies, Siegrist and Dragano [11] concluded that high occupational stress (as measured via a JDC or ERI imbalance) is associated with a 1.2 to 4.6-fold increased risk for depression. Additionally, occupational stress has been described to be the most important cause of burnout among adult workers [18], and seems to be closely associated with sleep complaints [19]. 

In the present study, we specifically focused on the occupation of law enforcement. Policing has often been described as a particularly (emotionally) stressful occupation [20,21]. In their professional lives, police officers have to deal with death, suffering, poverty, and physical threats; consequently, they are frequently required to draw on their physical and mental resources [22]. As emphasized by Waters et al. [23], some police officers start their career healthy, but develop severe health issues if the cumulative impact of stress becomes too big and takes its toll. Habersaat et al. [24] point out that police officers are a heterogeneous population, and that their health risk seems to depend more on individual perceptions of their work conditions than on characteristics of the division-specific work environment. This is in line with study findings reported by Gerber et al. [20], who observed that officers’ mental health is more closely related to their subjective stress perception than their shift work status, a factor associated with ill-health in other studies [25]. While ample evidence exists showing that chronic stress exposure contributes to both mental [26] and physical health issues [27,28], research is equivocal regarding the question of whether police officers are more at-risk for impaired health than other professions. For instance, previous research has shown that in the United States, many police officers have below-average fitness levels [29], and are more likely to be obese, to suffer from metabolic syndrome and to have higher total cholesterol levels than the general population [30]. By contrast, van der Velden et al. [31] reported that although policing is generally considered as a high-risk profession, no evidence exists to suggest that Dutch police officers are more likely to develop mental health disturbances than employees of banks, supermarkets, and psychiatric hospitals.

Given the negative health consequences associated with chronic occupational stress in both police officers and other professions [10,32], creating less stressful work environments and finding ways to make employees more resilient against stress are two major public health concerns [33,34]. Since the early 1980s, researchers have discussed whether participation in regular physical activity or having sufficiently high fitness levels might protect employees from the health hazards associated with chronic occupational stress [35]. While such “stress-buffering” effects of physical activity and physical fitness are supported in the literature [36], few studies have focussed on occupational stress. Among the existing studies, researchers often failed to use theory-based instruments to assess stress [37,38], focussed on physical demands at work instead of psychological strain [39], and relied on self-reports (instead of objective measures) to assess physical activity or physical fitness [40,41,42,43]. While the existing studies generally support the notion of physical activity/fitness as a stress-buffer, it must also be noted that few studies used objective measures to assess health. Nevertheless, in a 30-year follow-up study, Holtermann et al. [39] found that participants with high work demands had a higher risk of ischaemic heart disease and all-cause mortality, if compared to participants with low physical work demands. However, this only applied to men with low fitness levels, whereas no such relationship was found among men with high fitness. Similarly, Gerber et al. [37] reported that employees with high scores on a simple 1-item stress questionnaire had a higher total metabolic risk as compared to counterparts with lower stress levels. Again, this relationship was only observed among employees with low fitness levels, whereas no association between occupational stress and cardiometabolic risk occurred among employees with moderate or high fitness levels. Finally, Schmidt et al. [44] reported that employees with high scores on a maximal fitness test were less likely to be affected by high job-related self-control demands. 

Given this background, the purpose of the present study was to find out whether cardiorespiratory fitness levels moderated the relationship between occupational stress, cardiovascular and mental health outcomes. The present study goes beyond existing research in several ways: First, we used the two most established models (JDC/ERI) to assess occupational stress. Second, contrary to previous studies, in which researchers often used self-reported physical activity as an easily measurable proxy for physical fitness, we used an established submaximal fitness test to estimate participants’ maximal oxygen uptake. Third, contrary to most previous studies, we included objective physical health markers in order to find out whether the stress-buffering effects of physical fitness could be generalized beyond subjective health complaints and symptoms of mental ill-health. 

Based on the literature presented above, we formulated three hypotheses:Our first hypothesis was that higher levels of occupational stress will be associated with an increased cardiovascular risk, and more frequent mental health complaints, independent of the model used to operationalize occupational stress.Our second hypothesis was that higher levels of cardiorespiratory fitness will be associated with a lower cardiovascular risk, and fewer mental health complaints.Our third hypothesis was that cardiorespiratory fitness will moderate the relationship between occupational stress and physical/mental health indicators: Thus, the relationship will become smaller as a function of increasing fitness levels, independent on whether occupational stress is operationalized via the JDC or ERI model.

## 2. Materials and Methods 

### 2.1. Participants and Procedures

Participants were recruited from a police force in Basel, a bigger city in the northwestern, German-speaking part of Switzerland. All officers (*N* = 980, 290 females, 690 males) were invited to participate in the study, consisting of a comprehensive health check, including a cardiorespiratory fitness test, 7-day actigraphy, anthropometry, blood pressure assessment, a computerised cognitive test (facial emotion recognition), a functional movement screen, a lung function test, and an online-survey focusing on stress and mental health. Different channels were used to advertise the study including emails via intranet, video clips on the internal TV channel, printed flyers, and oral information during team meetings. For interested officers, more specific information (e.g., regarding the voluntary nature of participation, no negative consequences in case of non-participation, information about benefits and risk, information about measurements) was provided via short text modules and video messages via an e-learning tool. Data assessment took place between October 2017 and March 2018. As an incentive, each participant received a personalized health profile after the completion of the data assessment. Moreover, participants had the possibility to take part in a voluntary lifestyle coaching program.

No specific inclusion and exclusion criteria were applied. However, to be allowed to take part in the cardiorespiratory fitness test, participants were asked to sign the written informed consent and fill in an extended version of the Physical Activity Readiness Questionnaire (PAR-Q) [45]. Thus, workers were not allowed to perform the cardiorespiratory fitness test, if they reported any conditions that prevent participation in a submaximal fitness test, as defined by the American College of Sports Medicine (ACSM) [45]. In case of uncertainty, participants were asked to consult a general practitioner (and provide a doctor’s certificate). Furthermore, participants were only allowed to perform the cardiorespiratory fitness test if they did not report a temporary illness such as a cold or a fewer. If participants were not able/allowed to take part in the cardiorespiratory fitness test, they still could perform all other tests.

The data assessment took place during official working hours. The 120-min laboratory testing was carried out at the education and training center of the police force, and all tests were performed in a private setting in a specific room reserved across the entire study period by the same investigator.

The regional ethical review board (EKNZ: Project-ID: 2017-01477) approved the study, which was performed in accordance with the ethical principles laid down in the current edition of the Deceleration of Helsinki.

### 2.2. Measures

#### 2.2.1. Occupational Stress

Two different scales were used to assess work-related stress, referring to the two most widely used job-related stress theories [46], the Job Demand and Control (JDC) [47] and the Effort-Reward Imbalance (ERI) model [48]. The demand scale from the JDC model contains five items on a 4-point Likert-scale ranging from 1 (never) to 4 (often). For example, participants were asked whether their job requires them to work very fast, hard, or to accomplish large amounts of work. Participants also completed six items on the subscale pertaining to control (e.g., ‘I have freedom to make decisions about my job’). The items were summed to obtain subscale scores for job demand and job control. Because of the unequal number of items the JDC-ratio was calculated with the following formula: job demand/(job control × 0.8333). Values >1 of the JDC-ratio indicated stress with possible adverse health effects [49]. The validity and reliability of this instrument has been established previously [50]. The effort scale of the ERI model consists of five items anchored on a 5-point Likert-scale from 1 (none) to 5 (very high). Sample items were: ‘I have a lot of responsibility in my job’ or ‘I have many interruptions and disturbances in my job.’ The reward scale consists of 11 items with the same semantic anchors (e.g., ‘I receive the respect I deserve from my superior or a respective relevant person.’ or ‘Considering all my efforts and achievements, my job promotion prospects are adequate.’). The ERI-ratio was calculated with the following formula: effort/(reward × 0.4545). This measure has previously proven to be valid and reliable [51]. ERI-ratio scores above 1 reflect higher levels of job stress [48].

#### 2.2.2. Cardiovascular Risk Markers

Blood pressure, height, weight, waist circumference, body composition, blood lipids and blood glucose were assessed as cardiovascular risk markers. Blood pressure was measured with a portable device (OMRON M500, OMRON Healthcare Co. Ltd. 53, Kunotsubo, Terado-cho, Muko, Kyoto 617-0002 Japan), which was attached to the left arm. Two measurements were taken with approximately one minute’s break in between. The mean of the two measurements was used for further analyses [52]. Systolic and diastolic blood pressure were used as outcome measures. Height was measured to the nearest five mm using a stationary stadiometer. Weight and body composition were assessed with a wireless body composition monitor (BC-500, Tanita Corp., Tokyo, Japan). The participants were asked to fast three hours before the data assessment, to not take part in exhausting sport activities 24 h prior to the testing, to void their bladder prior to the data assessment, and to wear only light sport clothing (≤1 kg). The instrument assessed the participants’ weight to the nearest ten g, which was corrected for clothes (minus one kg). The Body Mass Index (BMI) was calculated (weight (kg) * height (m)^−2^) as an outcome variable. Moreover, body fat was assessed as an outcome variable. Blood lipids and blood glucose were assessed via finger prick, and instantly analyzed using the Alere Afinion AS100 Analyzer (Abbott Diagnostics, Alere Technologies, Rodeløkka NO-0504 Oslo, Norway). Alere lipid and glucose panel controls were used to ensure the reliability of the analyzers. Two drops of blood were needed to assess blood glucose (HbA1c), total cholesterol (TC), high-density lipoprotein cholesterol (HDL-C), low-density lipoprotein cholesterol (LDL-C), and triglycerides (TG). Capillary blood sampling is a frequently used minimal invasive and minimal painful method, and is regarded as being quicker and less distressing than venipuncture [53]. The finger prick was performed following the standardization protocol of the World Health Organization (WHO) [53], in order to maintain good clinical utility and high accuracy [52,54]. The validity and reliability of the applied instruments and procedures have been documented in previous research [55,56,57].

#### 2.2.3. Mental Health

We used the Shirom–Melamed Burnout Measure (SMBM) to assess occupational burnout symptoms [58]. The SMBM is a validated and widely used tool that comprises 14 items with response options on a 7-point Likert scale from one (almost never), to seven (almost always). The items can be assigned to three dimensions: six items refer to the aspects of physical fatigue, five items to cognitive weariness, and three items refer to emotional exhaustion. The mean score is built to obtain an overall burnout index, with values of ≥4.40 being considered as clinically relevant [59]. Subjective sleep complaints were assessed with the brief 7-item self-report Insomnia Severity Index (ISI) [60]. Referring to the last month, participants state their difficulties falling asleep, difficulties maintaining sleep, early morning awakenings, daytime fatigue, daytime performance, satisfaction with sleep, and worrying about sleep. Answers were given on a 5-point Likert scale, scored from 0 (e.g., no problem at all) to 4 (e.g., very severe problem), yielding a total of 0 to 28, with higher scores being indicative of more frequent sleep complaints. Scores should be interpreted as follows: 0–7 = absence of insomnia; 8–14 = sub-threshold insomnia; 15–21 = moderate insomnia; 22–28 = severe insomnia [61]. Adequate measurement properties have been shown previously [62]. The German version of the General Health Questionnaire (GHQ-12) was used to assess mental distress or minor psychiatric morbidities [61,63]. The GHQ-12 has been frequently used in the literature and its validity and reliability have been extensively reviewed [63]. Participants self-report their mental well-being during the previous week. Answers can be given on a 4-point Likert scale ranging with response options being 0 (not at all), 1 (same as usual), 2 (more than usual), and 3 (much more than usual). Sum scores range from 0 to 36, with higher scores reflecting higher levels of mental distress. Currently, no standard cut-off values exist for dividing up “cases” identified by a GHQ-12 score threshold. However, researchers have used the following categories to successfully establish links between the GHQ-12 and mortality (if response options 0 + 1 = 0, and 2 + 3 = 1): asymptomatic (0), subclinically symptomatic (1–3), symptomatic (4–6), highly symptomatic (7–12) [4,64]. In the present sample, internal consistency was acceptable across all mental health indicators. Cronbach’s alphas were 0.94 for burnout symptoms, 0.74 for sleep complaints, and 0.87 for mental distress.

#### 2.2.4. Cardiorespiratory Fitness

Cardiorespiratory fitness was tested with the Åstrand Fitness Test, a submaximal test performed on a bicycle ergometer [65]. Based on participants’ heart rate, maximal oxygen consumption (VO_2max_), which is a measure of cardiorespiratory fitness, was estimated indirectly [66]. Participants were instructed to avoid any strenuous activity for 24 h prior to the testing, as well as heavy meals, caffeine, or nicotine for three hours before the testing. Before starting the test, participants were equipped with a heart rate monitor. For men, the test protocol was standardized at a workload of 150 watts, for women at 100 watts. The workload was adjusted so that the heart rate is kept between 130 beats per minute (bpm) and 160 bpm, and between 120 bpm and 150 bpm in participants from the age of 40. The subjects were asked to cycle at a pace of 60 rotations per minute (rpm) for six minutes. At the end of each minute, the subject’s heart rate was noted. After two minutes, the participant reached a steady state, where only minor heart rate fluctuations occurred within the target heart rate zone. During the test, standardized encouragements were used and cancellation criteria were monitored. After six minutes, the test ended. Using a nomogram, the mean heart rate of minute five and six are compared against the watts that the participant trained at. Depending on the subject’s age, a correction factor was applied, and the VO_2max_ per kg was determined [67]. Gender and age-adjusted norms were used to classify participants in groups with low, moderate or high fitness levels [68].

### 2.3. Statistical Analyses

A power analysis (using G*Power 3.1.9.3, Heinrich Heine University, Dusseldorf, Germany) indicated that for F-tests (ANCOVA: Fixed effects, main effects and interactions) a minimum of 158 participants are needed to detect moderate effects (alpha error = 0.05, power = 0.80, df = 2, groups = 3).

Descriptive and inferential statistics were calculated using IBM SPSS Statistics 25 (IBM, Armonk, NY, USA). We carried out two separate two-factorial multivariate analyses of covariance (MANCOVAs) with (a) cardiovascular risk factors and (b) mental health outcomes as dependent variables, which were followed by a series of two-factorial analyses of covariance (ANCOVAs) to examine main and interaction effects of stress and CRF levels. To ensure a sufficient number of participants per cell, we compared two groups with low vs. high stress levels, and three groups with low, moderate, and high fitness levels. All analyses were controlled for age, gender, education (“What is your highest level of education; without the police training?”), employment rate (“What was your average employment level over the last 6 months?”), relationship status (“What is your current relationship situation?” Response options: single; married or in a relationship), children at home (“How many children do you have who still live at home?”), caregiving responsibility (“Are you currently the main person responsible for a family member or friend in need of care?”), supervisor status (“How many people do you have direct supervision responsibility for?”), work experience (“How many years have you been employed by the police?”), shift-work status (“Do you work in shifts or daytime?”), smoking (“Do you smoke?”), and medication intake (“Do you regularly take medication?”). To maximize statistical power, all outcomes were treated as continuous variables in the (M)ANCOVAs (and not as risk categories) [45]. Moreover, all missing values were replaced via expectation maximation (EM) algorithm [69]. The 5% level of significance was used to test main and interaction effects. Partial eta squared (*η*^2^) were used to interpret the strengths of the effects. Beyond single cardiometabolic risk factors, we also calculated a clustered metabolic risk index, which corresponded to a continuously distributed score (with higher scores being indicative of higher total cardiometabolic risk). To obtain this score, the z-standardized residuals were computed for the following variables [45]: (SBP/DBP)/2, BMI, waist circumference, body fat, TC, HDL-C (*−1), LDL-C, TG and Hb1Ac.

## 3. Results

### 3.1. Sample Description

Two hundred and one (201) officers (129 men, 72 women) volunteered to take part in the data assessment. The mean age was 38.6 years (standard deviation [SD] = 10.1). Twenty percent reported living alone, whereas 80% were married or living with someone. On average, officers were working since 12.9 years (SD = 8.8) in the police force (range: 1–37 years). Most of the participants were full-time employees (83%), 12% reported employment rates between 50 and 90%, and 6% reported an employment rate between 20 and 40%. Fourteen percent of the participants have higher education (college/university degree), 32% have completed high school or advanced vocational training, whereas 54% have basic vocational training. More than half of the participants (57%) had no children, whereas 16% had one child, 24% two children, and 3% three or more children. Three percent currently had carer responsibilities, 47% percent of the participants were shift-workers, 20% were smokers, and 16% reported that they take medication on a regular basis. Finally, 37% had supervisor status, with supervisors being responsible for 1 to 80 subordinate employees (M = 10.7, SD = 14.8). 

### 3.2. Descriptive Statistics and Correlations between Independent and Dependent Variables

Descriptive statistics for all independent and dependent variables are summarized in Table 1. Following Cifkova et al. [70], 17.9% and 32.2%, of the participants were classified as hypertensive based, respectively, on the cut-off for systolic (≥140 mmHg) and diastolic blood pressure (≥90 mmHg). Based on WHO [71] recommendations, 52.2% of the participants were classified as overweight (BMI ≥ 25.0 kg/m^2^), and 9.5% as obese (BMI ≥ 30.0 kg/m^2^). Following the expert panel of the National Cholesterol Education Program [72], which defined a waist circumference of ≥80 cm (women) and ≥94 cm (men) as risk factor for metabolic syndrome, 65.3% of women and 45.7% of men exceeded this cut-off. With regard to body fat, the WHO [71] recommends maximum levels of ≥32% for women and ≥25% for men. Based on these standards, 22.2% of women and 11.1% of men had excessive body fat levels. Following Rodondi et al. [73], the following clinically relevant cut-offs should be considered for total cholesterol (≥5.6 mmol/L), HDL-C (≤1.41 mmol/L) and LDL-C (≥3.40 mmol/L). In the present sample, these cut-offs were exceeded by 25.9% (total cholesterol), 16.4% (HDL-C) and 10.9% (LDL-C) of the participants. Moreover, based on the recommendations of the expert panel of the National Cholesterol Education Program [72], 31.3% exceeded the clinical cut-off for triglycerides (≥1.71 mmol/L). Following the criteria of the American Diabetes Association [74], we identified 31 participants with prediabetes (HbA1c ≥ 5.7%) and three participants with diabetes (HbA1c ≥ 6.5%). 

With regard to mental health, burnout scores of ≥4.40 can be considered as clinically relevant [59], whereas scores of ≥15 [61] and 4 [75], respectively, point towards clinical levels of insomnia or excessive mental distress. Based on these standards, 4.0%, 6.5% and 14.4% reported clinically relevant levels of burnout, sleep complaints, and excessive mental distress.

Two of the variables (triglycerides, HbA1c) exceeded the recommended values for skewness (value of 2) and kurtosis (value of 7) [76]. We therefore built the logarithm of these scores before calculating the ANCOVAs.

With regard to stress, 80 participants (40%) had JDC scores >1 and 54 participants (27%) had ERI scores >1. These participants were considered as being exposed to high occupational stress levels. Finally, based on the ACSM classification standards, 53 participants (26%) had low CRF, 59 participants (29%) had moderate CRF, and 89 participants (44%) had high CRF levels.

Bivariate correlations between independent and dependent study variables are displayed in Table 1. Significant correlations were found between both occupational stress measures and the three mental health indicators. Few significant correlations existed between the JDC and ERI ratio and the cardiovascular health outcomes. The only significant correlations were found between the JDC ratio and HbA1c, and the ERI ratio and BMI, waist circumference, as well as the total cardiometabolic risk factor score. Higher CRF was correlated with higher BMI, higher waist circumference, lower body fat, and lower HDL cholesterol. By contrast, no significant correlations were found between participants’ CRF level and any of the mental health indicators. Weak to moderate correlations existed between most of the various cardiometabolic risk factors, whereas moderate to strong correlations were observed between burnout symptoms, insomnia symptoms and overall mental wellbeing.

### 3.3. Main and Interaction Effects

Job demand and control (JDC model). If stress was operationalized via the JDC model, the MANCOVA showed a significant main effect for the factor CRF, F(20, 350) = 3.2, *p* < 0.001, *η*^2^ = 0.154, whereas no significant effects occurred for the factor stress, F(10, 175) = 0.4, *p* = not significant (ns), *η*^2^ = 0.023, or the interaction between CRF and stress, F(20, 350) = 1.4, *p* = ns, *η*^2^ = 0.075, after controlling for covariates. Table 2 further shows the findings of the ANCOVAs, displaying the means and standard deviations for the six subgroups with low vs. high occupational stress, and low, moderate vs. high cardiorespiratory fitness. These values are followed by the inferential statistics (F, *η*^2^ values) for the main effects of stress and cardiovascular fitness, and the interaction between these two constructs. Table 2 yields a relatively consistent picture across all outcomes. Thus, participants with higher fitness levels showed more favorable scores across almost all cardiovascular risk factors (BMI, waist circumference, body fat, HDL cholesterol, triglycerides, HbA1c). No significant main effects for cardiorespiratory fitness were found for systolic and diastolic blood pressure, total cholesterol, and LDL cholesterol. In line with these findings, a relatively strong main effect (12.7% of explained variance) was found for the total cardiometabolic risk index, showing that participants with the highest fitness level had the most favorable scores, followed by those with moderate and low fitness levels (see Table 2).

With regard to the mental health indicators, the MANCOVA yielded a significant main effect for the factor stress, F(3, 182) = 6.8, *p* < 0.001, *η*^2^ = 0.101, whereas no significant effects were identified for CRF, F(6, 364) = 0.6, *p* = ns, *η*^2^ = 0.009, and the interaction between CRF and stress, F(6, 364) = 1.0, *p* = ns, *η*^2^ = 0.017. In line with this finding, we found three significant main effects in the univariate ANCOVAs for occupational stress, showing that independent of their fitness levels, officers who perceive higher levels of occupational stress report more symptoms of mental ill-health than their less stressed colleagues. In the univariate analyses, we did not find any significant interaction effects between occupational stress and cardiorespiratory fitness for any of the study variables.

Effort-reward imbalance (ERI model). If stress was operationalized via the ERI model, the MANCOVA showed a significant main effect for the factor CRF, F(20, 350) = 3.2, *p* < 0.001, *η*^2^ = 0.155, whereas no significant effects occurred for the factor stress, F(10, 175) = 0.9, *p* = ns, *η*^2^ = 0.047, or the interaction between CRF and stress, F(20, 350) = 1.1, *p* = ns, *η*^2^ = 0.062, after controlling for covariates. As shown in Table 3, the results pattern described for the JDC model was similar when we carried out the univariate ANCOVAs with the ERI model. Again, we found significant main effects for fitness for the following variables: DBP, BMI, waist circumference, body fat, HDL cholesterol, triglycerides, and total cardiometabolic risk. By contrast, no significant main effects were found for occupational stress. Finally, contrary to the results of the multivariate analysis, three significant univariate interactions were observed for total cholesterol, triglycerides, and total cardiometabolic risk. As shown in Figure 1 for the total cardiometabolic risk index, participants with low CRF levels had particularly high cardiometabolic risk scores if exposed to high occupational stress.

As for the JDC model, in the MANCOVA with the mental health indicators as outcomes, we found a significant main effect for stress F(3, 191) = 5.6, *p* < 0.001, *η*^2^ = 0.081, whereas no significant effects occurred for CRF F(6, 382) = 1.2, *p* = ns, *η*^2^ = 0.018, and the interaction between stress and CRF, F(6, 382) = 0.6, *p* = ns, *η*^2^ = 0.009, if stress was operationalized via an imbalance between job-related efforts and reward. In line with this, the univariate ANCOVAs indicated significant main effects across all mental health indicators, showing that officers with higher stress levels report more mental health complaints. The effect sizes (4.9 to 10.0% of explained variance) were moderate to strong. No significant interaction effects were found between occupational stress and cardiorespiratory fitness.

## 4. Discussion

The key findings of the present study are that high cardiorespiratory fitness levels were associated with a reduced cardiometabolic risk, whereas high levels of occupational stress were associated with more burnout symptoms, sleep complaints and increased overall psychological distress. However, moderation effects of cardiorespiratory fitness on the relationship between occupational stress and the examined health outcomes were only found for the ERI ratio on total cholesterol, triglycerides and the total cardiometabolic risk index. The interaction revealed that participants with low fitness levels had a particularly high cardiometabolic risk if they were exposed to high levels of this type of occupational stress.

Our first hypothesis was that higher occupational stress would be associated with an increased cardiovascular risk, as well as decreased mental health. While the association with mental health was confirmed, in the (M)ANCOVAS, no significant differences were found between officers with low versus high occupational stress levels with regard to cardiovascular risk factors. This pattern holds true for both occupational stress models. 

The fact that occupational stress was associated with mental health complaints is in line with previous research [11]. For instance, high job strain and high effort-reward imbalance predisposed the development of depression, burnout, and general mental health (GHQ-12) in prospective cohort studies with 2555 dentists [77] and 22,899 public service workers [78]. A close association between occupational stress and mental health has also been observed in police officers [26,79]. By contrast, the fact that occupational stress was unrelated to further health outcomes in the present sample was unexpected and at odds with previous research among police officers [28]. However, the existing findings are difficult to compare as researchers have used various instruments to assess occupational stress. In this context, Magnavita et al. [28] criticized that “most of the studies available provided a general non-specific measure of perceived stress that included both general and occupational stress” (p. 386).

Our second hypothesis was that higher levels of cardiorespiratory fitness would be associated with a lower cardiovascular risk, and fewer mental health complaints. Again, partial support was found for this hypothesis. Thus, in line with previous research in the police setting [80], we observed a favorable pattern of cardiovascular health markers among police officers with higher cardiovascular fitness levels. By contrast, no significant associations were found between officers’ fitness levels and their mental health. Our findings were in line with previous studies showing that cardiorespiratory fitness had the potential to enhance people’s cardiovascular health [81]. In a recent work by the American Heart Association, Ross et al. [82] provided a review of the rich body of evidence linking higher cardiovascular fitness to healthier levels of blood pressure, visceral adiposity, insulin sensitivity, as well as lipid and lipoprotein profiles. In a cross-sectional study with 2368 men and 2263 women, Aspenes et al. [83] measured cardiorespiratory fitness by spiroergometry on a treadmill. Despite the generally low fitness levels identified in their study population, logistic regression revealed that with every 5-mL·kg^−1^·min^−1^ decline in VO_2max_ the odds ratio for the cardiovascular risk factor cluster increased by 56% [83]. Fernström et al. [84] reported similar findings among young adults aged 18 to 25 years.

With regard to mental health, positive associations with cardiorespiratory fitness have been reported previously. For instance, Vancampfort et al. [85] reported that people with severe mental illness have significantly lower cardiorespiratory fitness levels than healthy controls. Gerber et al. [86] further found that cardiorespiratory fitness was associated with burnout symptoms among Swedish health care workers, while Ortega et al. [87] found cardiorespiratory fitness and mental well-being to be independent predictors of mortality in a prospective study of 4888 participants. Thus, the fact that cardiorespiratory fitness and mental health were not associated with each other in the present sample was unexpected. However, the absence of this relationship can be attributed to the fact that our sample might have been healthier than those in prior research, entailing a possible ceiling effect [88]. Thus, according to ACSM standards, only 26% had poor or very poor fitness levels, whereas 44.5% had fair to excellent, and 30% even superior fitness levels. Moreover, only 4 to 7% exceeded the clinical threshold for burnout and sleep complaints, respectively, which is considerably below the rates found in previous studies [37,62].

Our third hypothesis was that cardiorespiratory fitness would moderate the relationship between occupational stress and physical/mental health indicators, but our findings only partly supported this notion. When stress was operationalized via the JDC model, no significant moderation was found. Nevertheless, when stress was operationalized via the ERI model, a significant interaction was found between occupational stress and CRF on total cholesterol, triglycerides and total cardiometabolic risk, showing that participants with low CRF had particularly high cardiometabolic risk scores when they perceived a high imbalance between efforts and reward. This finding is in line with two studies with Swedish health care workers [37] and Swiss schoolchildren [89], where similar interactions were observed. As shown in Table 1, only a few (weak) correlations were found between work-related stress on the cardiovascular health indicators. Accordingly, the potential for CRF to moderate this relationship was limited in the present sample. This may also explain why significant interaction effects were found for some, but not all cardiovascular health indicators. Furthermore, the correlations between the JDC and ERI index with the cardiovascular health outcomes pointed into different directions. This may be the reason why interaction effects were only found for the ERI ratio, whereas no such effects were observed for the JDC ratio. In a recent review, Klaperski [90] showed that physical fitness buffered the negative effects of stress in the majority of the eligible studies. However, one study cited in this review found no support for a stress-buffering effect on blood pressure and HDL-C in law enforcement officers [91]. Young [91] attributed these results to the chronicity of the measure of work stress, which might additionally not have been sensitive enough to discern physiological variations in healthy individuals. Some authors further state that physical fitness possibly buffers the acute stress response in mentally demanding occupations, rather than the long-term effects of stress [92]. Although well established in the laboratory setting [93], to date only a few studies investigated the stress-buffering effects of physical fitness on acute stress reactivity in real life. In a randomized controlled trial, von Haaren et al. [33] showed that a 20-week aerobic exercise training buffered the physiological stress response to real life stressors. The authors were able to show associations of an improved cardiorespiratory fitness and ambulatory assessed heart rate variability during two days of the examination period in 61 previously sedentary university students. Due to the current relevance of occupational stress, there is a clear demand for further evidence on this issue.

The strengths of our study are that we assessed occupational stress with standardized and validated instruments that are based on the two of the most popular and internationally accepted stress models [28]. Using an established cut-off to distinguish between participants with low versus high stress levels further allows a comparison with existing studies. Moreover, we assessed a broad range of health indicators to provide a comprehensive picture of the officers’ health status. Another advantage was that cardiorespiratory fitness was assessed objectively, and that age- and gender-adjusted norms were used to classify participants in groups with low, moderate or high fitness levels. Finally, this is one of the few studies in which fitness-based stress-buffer effects were tested with objective health outcomes. This is important, as most previous studies have focused on (subjectively reported) health outcomes. Thus, in these studies the relationship between occupational stress and health might be inflated because of the shared method variance.

By contrast, some aspects limit the generalizability of our data. First, our study is cross-sectional, which precludes conclusions about causality [28,92]. Second, several blood lipid measures were considered when we calculated the overall cardiometabolic risk index [45]. However, as blood lipids were only weakly associated with cardiorespiratory fitness, it can be assumed that using our formula is likely to have resulted in an under-estimation of the association between cardiorespiratory fitness and overall cardiometabolic risk. Third, given the voluntary nature of the study, a recruitment bias is conceivable. However, the fitness categories still discriminated between participants with more or less favorable cardiometabolic risk patterns, and the ERI/JDC categories explained significant amounts of variance in the mental health outcomes. Fourth and last, physical work place characteristics were not systematically assessed in the present study [94]. However, information regarding participants’ shift work status allowed us to control for the fact of whether officers engaged in office or field service.

## 5. Conclusions

Our findings suggest that occupational stress is negatively associated with police officers’ mental wellbeing. Our results also suggest that occupational stress may be more closely related to psychological health than physical health. Therefore, work health programs for police officers should consider the early screening of burnout, sleep disturbances, and overall mental well-being. CRF, as measured with a submaximal bicycle test, appears to be a relevant marker for cardiovascular health in police officers. Moreover, officers with moderate or high CRF levels have a lower overall cardiometabolic risk when they perceive a strong imbalance between their invested efforts and their reward at work. To increase cardiovascular health, we encourage the inclusion of fitness tests in routine health checks, and the promotion of physical activity to further increase the fitness levels among police officers.

## Figures and Tables

**Figure 1 ijerph-16-02349-f001:**
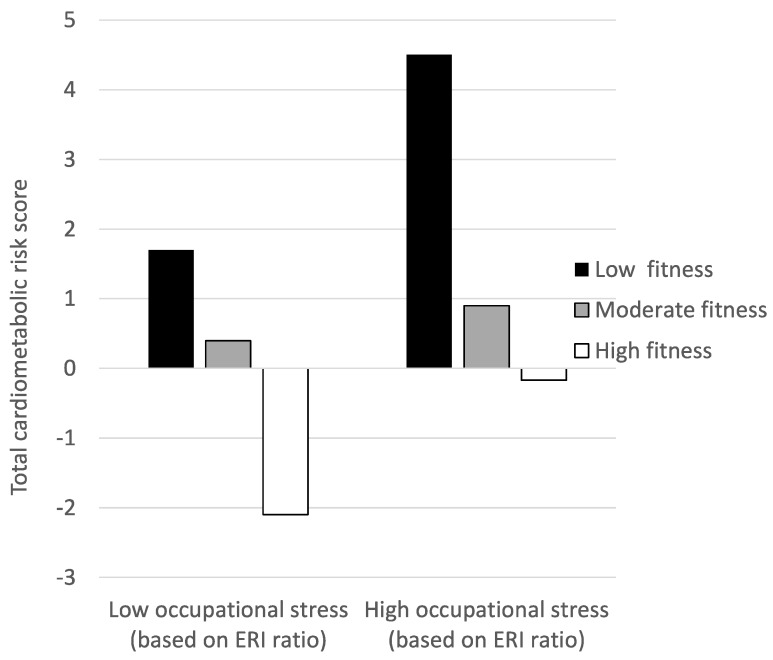
Graphical representation of the interaction between occupational stress (as assessed with the ERI ration) and cardiorespiratory fitness on total cardiometabolic risk. Note: Higher scores on the total cardiometabolic risk index reflect higher cardiometabolic risk.

**Table 1 ijerph-16-02349-t001:** Descriptive statistics for and bivariate correlations between independent and dependent variables.

Descriptive Statistics	*n*	M	SD	Range	Skew	Kurt	Bivariate Correlations between Independent and Dependent Variables															
							1.	2.	3.	4.	5.	6.	7.	8.	9.	10.	11.	12.	13.	14.	15.	16.
**Stress**																						
1. JDC ratio	201	1.0	0.2	0.5 to 1.6	0.7	0.4	-															
2. ERI ratio	201	0.9	0.3	0.3 to 2.0	0.9	1.5	0.33 *	-														
**Cardiorespiratory fitness**																						
3. VO_2max_	201	3.5	0.9	1.6 to 5.9	0.3	−0.6	0.10	0.05	-													
**Cardiovascular risk factors**																						
4. SBP (mmHg)	201	129	13	104 to 172	0.5	0.2	−0.05	0.13	0.10	**-**												
5. DBP (mmHg)	201	85	10	63 to 118	0.2	−0.3	−0.11	0.12	0.02	0.84 *	**-**											
6. BMI (kg·m^−2^)	201	25.8	3.6	17.9 to 37.4	0.7	0.8	0.00	0.16 *	0.25 *	0.33 *	0.37 *	**-**										
7. Waist circumference (cm)	201	91.1	11.3	60.0 to 126.0	0.4	0.8	−0.08	0.14 *	0.20 *	0.37 *	0.43 *	0.83 *	**-**									
8. Body fat (%)	201	21.8	7.3	5.6 to 42.6	0.4	−0.2	−0.01	0.02	−0.38 *	0.01	0.03	0.37 *	0.30 *	**-**								
9. TC (mmol·L^−1^)	201	5.0	1.0	2.9 to 10.4	1.1	3.2	−0.11	0.05	0.00	0.28 *	0.34 *	0.31 *	0.32 *	0.14	**-**							
10. HDL-C (mmol·L^−1^)	201	1.8	0.4	0.6 to 2.6	0.1	−0.5	−0.07	−0.08	−0.14 *	−0.13	−0.15 *	−0.35 *	−0.35 *	0.15 *	0.20 *	**-**						
11. LDL-C (mmol·L^−1^)	201	2.5	0.8	1.0 to 5.0	0.6	0.2	−0.06	0.05	0.05	0.24 *	0.33 *	0.28 *	0.29 *	0.05	0.81 *	−0.06	**-**					
12. TG (mmol·L^−1^)	201	1.7	1.2	0.5 to 7.4	2.3 (0.6)	6.5 (0.2)	−0.07	0.09	0.03	0.26 *	0.27 *	0.42 *	0.41 *	0.05	0.53 *	−0.29 *	0.12	**-**				
13. HbA1c (%)	201	5.4	0.3	4.9 to 7.5	2.4 (1.7)	13.7 (6.9)	−0.15 *	0.03	−0.06	0.19 *	0.26 *	0.27 *	0.35 *	0.09	0.09	−0.24 *	0.14 *	0.14	**-**			
14. Total cardiometabolic risk	201	0.0	4.7	−10.6 to 16.7	0.7	0.8	−0.10	0.15 *	0.06	0.63 *	0.69 *	0.78 *	0.80 *	0.32 *	0.62 *	−0.38 *	0.57 *	0.60 *	0.47 *	**-**		
**Mental health indicators**																						
15. Burnout symptoms	201	2.5	1.0	1 to 6	0.9	0.7	0.31 *	0.31 *	−0.08	−0.08	−0.05	0.02	−0.02	0.14 *	−0.08	0.03	−0.09	−0.04	−0.03	−0.05	**-**	
16. Insomnia symptoms	201	7.9	4.3	0 to 22	0.5	0.0	0.23 *	0.21 *	−0.06	−0.06	−0.05	0.00	−0.02	0.05	−0.01	−0.08	0.00	0.03	−0.04	0.00	0.46 *	**-**
17. Overall mental wellbeing	201	1.7	2.5	0 to 11	1.7	2.2	0.26 *	0.25 *	−0.04	0.02	0.00	0.14 *	0.09	0.18 *	−0.02	0.08	−0.08	0.02	−0.04	0.04	−0.62 *	0.31 *

Notes: JDC = Job Demands and Control; ERI = Effort Reward Imbalance; VO_2max_ = Maximal Oxygen Uptake; SBP = Systolic Blood Pressure; DBP = Diastolic Blood Pressure; BMI = Body Mass Index; TC = Total Cholesterol; HDL-C = High-Density Lipoprotein Cholesterol; LDL-C = Low-Density Lipoprotein Cholesterol; HbA1c = Glycated Hemoglobin; M = Mean; SD = Standard Deviation; Skew = Skewness; Kurt = Kurtosis. Values in brackets for TG and HbA1c for skewness and kurtosis represent values after building the logarithm. * *p* < 0.05.

**Table 2 ijerph-16-02349-t002:** Differences in cardiometabolic risk factors, and mental health indicators, dependent on participants’ levels of CRF and perceived occupational stress (as assessed via the job demand and control [JDC] score).

	Low Occupational Stress (JDC Score)			High Occupational Stress (JDC Score)			Stress (JDC)		CRF		Stress (JDC) × CRF	
	Low CRF (*n* = 36)	Moderate CRF (*n* = 34)	High CRF (*n* = 51)	Low CRF (*n* = 17)	Moderate CRF (*n* = 25)	High CRF (*n* = 38)						
	M ± SD	M ± SD	M ± SD	M ± SD	M ± SD	M ± SD	F	*η^2^*	F	*η^2^*	F	*η^2^*
**Cardiovascular risk factors**												
SBP (mmHg)	129 ± 12	130 ± 8	129 ± 15	132 ± 14	132 ± 15	127 ± 13	1.6	0.009	1.2	0.013	3.0	0.032
DBP (mmHg)	88 ± 11	86 ± 7	83 ± 10	86 ± 12	86 ± 10	83 ± 11	0.5	0.002	2.9	0.031	0.7	0.007
BMI (kg·m^−2^)	27.8 ± 4.7	26.3 ± 3.3	24.0 ± 2.7	26.6 ± 4.0	25.8 ± 3.4	25.8 ± 2.6	0.0	0.000	7.1 **	0.071	1.6	0.017
Waist circumference (cm)	97.6 ± 13.6	93.9 ± 10.9	85.6 ± 9.0	95.1 ± 12.3	90.4 ± 9.3	88.6 ± 8.9	0.0	0.000	11.8 ***	0.114	0.9	0.010
Body fat (%)	24.9 ± 7.7	20.8 ± 5.5	20.4 ± 7.3	23.6 ± 6.2	23.4 ± 8.2	19.8 ± 7.0	0.5	0.003	22.1 ***	0.194	1.4	0.015
TC (mmol·L^−1^)	5.1 ± 1.4	5.1 ± 1.1	5.1 ± 0.9	5.3 ± 1.2	4.9 ± 0.8	4.8 ± 0.8	0.2	0.000	0.7	0.007	1.1	0.012
HDL-C (mmol·L^−1^)	1.7 ± 0.4	1.8 ± 0.4	2.0 ± 0.4	1.6 ± 0.5	1.8 ± 0.5	1.9 ± 0.3	0.2	0.001	5.5 **	0.056	0.0	0.000
LDL-C (mmol·L^−1^)	2.5 ± 1.0	2.4 ± 0.8	2.5 ± 0.8	2.6 ± 0.8	2.4 ± 0.7	2.3 ± 0.6	0.0	0.000	0.6	0.007	0.7	0.007
TG (mmol·L^−1^)	1.9 ± 1.3	2.0 ± 1.5	1.4 ± 0.6	2.3 ± 1.9	1.6 ± 1.0	1.4 ± 0.7	0.1	0.001	3.7 *	0.038	1.9	0.020
HbA1c (%)	5.5 ± 0.3	5.5 ± 0.4	5.4 ± 0.2	5.5 ± 0.3	5.5 ± 0.2	5.4 ± 0.2	0.0	0.000	3.6 *	0.038	0.2	0.003
Total cardiometabolic risk	2.8 ± 7.7	0.8 ± 5.0	−2.0 ± 4.9	2.4 ± 6.7	0.3 ± 5.2	−2.0 ± 4.0	0.5	0.003	13.3 ***	0.127	0.2	0.002
**Mental health indicators**												
Burnout symptoms	2.3 ± 0.9	2.4 ± 1.1	2.4 ± 0.8	2.6 ± 0.9	2.9 ± 1.4	2.9 ± 1.0	15.9 ***	0.080	1.0	0.010	1.8	0.020
Sleep complaints	7.7 ± 4.1	7.2 ± 3.5	6.4 ± 3.6	8.8 ± 4.3	9.4 ± 5.2	9.1 ± 5.0	6.5 *	0.034	0.6	0.007	1.6	0.017
Overall mental distress	1.4 ± 2.0	1.1 ± 2.0	0.9 ± 1.8	1.8 ± 2.1	2.9 ± 3.3	2.5 ± 3.2	14.1 ***	0.071	1.0	0.010	1.2	0.013

Notes: SBP = Systolic Blood Pressure; DBP = Diastolic Blood Pressure; BMI = Body Mass Index; TC = Total Cholesterol; HDL-C = High-Density Lipoprotein Cholesterol; LDL-C = Low-Density Lipoprotein Cholesterol; TG = Triglycerides; HbA1c = Glycated Hemoglobin; JDC = Job Demands and Control; CRF = Cardiorespiratory Fitness; M = Mean; SD = Standard Deviation. All analyses controlled for age, gender, education, employment rate, relationship status, children at home, caregiving responsibility, supervisor status, shift-work status, medication intake. * *p* < 0.05. ** *p* < 0.01. *** *p* < 0.001.

**Table 3 ijerph-16-02349-t003:** Differences in cardiometabolic risk factors, and mental health indicators, dependent on participants’ levels of CRF and perceived occupational stress (as assessed via the effort-reward imbalance [ERI] score).

	Low Occupational Stress (ERI Score)			High Occupational Stress (ERI Score)			Stress (ERI)		CRF		Stress (ERI) × CRF	
	Low CRF (*n* = 34)	Moderate CRF (*n* = 44)	High CRF (*n* = 69)	Low CRF (*n* = 19)	Moderate CRF (*n* = 15)	High CRF (*n* = 20)						
	M ± SD	M ± SD	M ± SD	M ± SD	M ± SD	M ± SD	F	*η* *^2^*	F	*η* *^2^*	F	*η* *^2^*
**Cardiovascular risk factors**												
SBP (mmHg)	130 ± 13	130 ± 11	129 ± 15	131 ± 12	132 ± 15	127 ± 13	0.1	0.001	1.9	0.020	1.8	0.019
DBP (mmHg)	87 ± 11	85 ± 8	83 ± 11	87 ± 12	87 ± 10	83 ± 7	0.2	00.001	4.1 *	0.043	1.3	0.014
BMI (kg·m^−2^)	27.0 ± 3.8	25.9 ± 3.4	24.4 ± 2.8	28.1 ± 5.6	26.8 ± 3.0	25.2 ± 2.4	1.7	0.009	9.7 ***	0.095	1.1	0.012
Waist circumference (cm)	95.5 ± 11.8	92.5 ± 10.5	86.2 ± 8.8	99.2 ± 15.3	92.2 ± 10.1	89.3 ± 9.3	0.2	0.001	12.6 ***	0.121	0.4	0.004
Body fat (%)	24.4 ± 8.0	20.0 ± 6.1	20.7 ± 7.0	24.6 ± 6.0	24.4 ± 8.5	18.4 ± 8.2	0.7	0.004	21.2 ***	0.187	0.1	0.002
TC (mmol·L^−1^)	4.9 ± 1.0	5.0 ± 1.0	5.0 ± 0.8	5.6 ± 1.7	5.2 ± 0.9	5.0 ± 0.9	2.4	0.013	1.3	0.014	3.0 *	0.032
HDL-C (mmol·L^−1^)	1.7 ± 0.4	1.7 ± 0.4	1.9 ± 0.4	1.7 ± 0.5	1.9 ± 0.4	1.9 ± 0.4	0.7	0.004	5.5 **	0.057	0.1	0.001
LDL-C (mmol·L^−1^)	2.5 ± 0.9	2.4 ± 0.7	2.4 ± 0.7	2.7 ± 1.0	2.4 ± 0.8	2.5 ± 0.7	0.1	0.000	0.9	0.011	0.8	0.009
TG (mmol·L^−1^)	1.7 ± 0.9	1.7 ± 1.3	1.4 ± 0.6	2.5 ± 2.2	2.0 ± 1.4	1.4 ± 0.8	3.4	0.018	4.9 **	0.050	3.6 *	0.038
HbA1c (%)	5.5 ± 0.3	5.5 ± 0.4	5.4 ± 0.2	5.5 ± 0.4	5.5 ± 0.2	5.4 ± 0.2	1.2	0.007	2.9	0.031	0.6	0.007
Total cardiometabolic risk	1.7 ± 5.9	0.4 ± 5.1	−2.1 ± 4.5	4.5 ± 9.3	0.9 ± 5.2	−1.7 ± 4.5	0.5	0.003	16.8 ***	0.154	3.1 *	0.033
**Mental health indicators**												
Burnout symptoms	2.3 ± 0.9	2.4 ± 1.1	2.4 ± 0.8	2.6 ± 0.9	3.1 ± 1.2	3.0 ± 1.2	14.1 ***	0.071	2.2	0.023	1.5	0.016
Sleep complaints	7.7 ± 4.1	7.2 ± 3.5	6.4 ± 3.6	8.8 ± 4.3	9.4 ± 5.2	9.1 ± 5.0	10.0 **	0.049	0.6	0.007	1.0	0.010
Overall mental distress	1.4 ± 2.0	1.1 ± 2.0	0.9 ± 1.8	1.8 ± 2.1	2.9 ± 3.3	2.5 ± 3.2	17.1 ***	0.081	1.1	0.011	1.0	0.011

Notes: SBP = Systolic Blood Pressure; DBP = Diastolic Blood Pressure; BMI = Body Mass Index; TC = Total Cholesterol; HDL = High Density Lipoproteins; LDL = Low Density Lipoproteins; TG = Triglycerides; HbA1c = Glycated Hemoglobin; ERI = Effort Reward Imbalance; CRF = Cardiorespiratory Fitness, M = Mean; SD = Standard Deviation. All analyses controlled for age, gender, education, employment rate, relationship status, children at home, caregiving responsibility, supervisor status, shift-work status, and medication intake. * *p* < 0.05. ** *p* < 0.01. *** *p* < 0.001.

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
