# Peer review of "Does Cardiorespiratory Fitness Moderate the Association between Occupational Stress, Cardiovascular Risk, and Mental Health in Police Officers?"

_ijerph, 2019, doi:10.3390/ijerph16132349_

Round 1

Reviewer 1 Report

This cross-sectional study by Schilling et al. examined whether cardiorespiratory fitness (CRF) would modulate the association between occupational stress, cardiovascular risk and mental health in 201 Police Officers (included 36% women). The authors concluded that CRF was associated with reductions in cardiometabolic risk factors, whereas high occupational stress was associated with more burnout symptoms, sleep complaints and an overall increase in psychological distress. However, there were no effects of CRF on the relationship between occupational stress and the examined health outcomes. Overall, the study appears to be appropriately done and written carefully. However, there are several concerns to this manuscript.

Although there were no modulatory effects of CRF on the relationship between occupational stress and the examined health outcomes, it is unclear why the authors described this statement in their conclusion: “to increase cardiovascular health, including fitness tests in routine health checks and promoting physical activity to further increase cardiorespiratory fitness appears worthwhile.” The conclusion should be modified based on the results from this study.

Personally, I think this manuscript would benefit from a mediation analysis. Furthermore, the authors should consider examining the cross-sectional associations of occupational stress and health outcomes with CRF, and additionally, analyzing if CRF modulated these associations.

Page 3, lines 46-47, the authors stated: “if participants were not able/allowed to take part in the CRF test, they still could perform all other tests.” It is not clear whether all subjects did all tests. If not, please provide the number of participants on each variable in table 1.

How many hours were the participants fasted because fasting blood was drawn? At least three hours? If so, is this shorter duration of fasting long enough to be valid for an accurate analysis of lipid profiles? In addition, the authors should provide the test-retest reproducibility of blood sample using finger prick. How did the authors analyze the blood samples (i.e. HbA1c, lipid profiles)?

In the statistical analysis, the authors should provide the correlations of maximal oxygen uptake with all variables; occupational stress variables (JDC, ERI), cardiovascular risk factors and mental health outcomes (burnout symptoms, sleep complaints) in one table. What kind statistic program did the authors use in the present study?

In Tables 2 and 3, why did the authors control for several separate confounders: (a) age and gender as well as (b) age, gender and employment rate……..(i)? These confounders should be similar controlled for each dependent variable.  

There are major inconsistencies in how the results were reported in the results section compared with the results presented in the Tables. It is imperative that the description of the results section be consistent with the numerical results in Tables 1-3. In addition, please provide information about SES, HTN, obesity status, etc. that were described the results section into Table 1. Finally, please include information on JDC, ERI score, fitness levels, etc. into Table 1.

In line 16 of the results section, the authors stated “see Figure 1,” but there was no Figure 1 in this present paper.

Please provide more detailed information about the characteristics of the participants into table 1.

The authors did not really touch on sex differences at all in the results or discussion. Minimally, the authors should update their results presented in Tables 2 and 3 to be separated by sex, in additional to those for the entire sample.

Nobody had diabetes? Any subject used medications which may affect mental or physical health, such as anti-depressive drugs, NSAIDS or antioxidant? Another limitation to be noted is the lack of adjustment for socioeconomic status.

I don’t know what you mean by “moderation effects”

Author Response

Basel, 27 June 2019

Dear Prof. Weldon Luo,

We refer to your e-mail dated 17 June 2019 and thank you and the two external referees very much indeed for the opportunity to submit a revision of the aforementioned manuscript for consideration for publication in the International Journal of Environmental Research and Public Health. We found the comments of the reviewers very helpful and constructive in crafting a revision.

As per your request, we are submitting the document electronically. Below, we respond to each point raised by the reviewers. Please note that in the edited manuscript, we have highlighted all changes using a yellow marker. If we can provide any additional information, or make any additional changes, please do not hesitate to let us know.

Sincerely,

René Schilling (on behalf of all co-authors)

Reviewer #1

********************************************************************************************************************

This cross-sectional study by Schilling et al. examined whether cardiorespiratory fitness (CRF) would modulate the association between occupational stress, cardiovascular risk and mental health in 201 Police Officers (included 36% women). The authors concluded that CRF was associated with reductions in cardiometabolic risk factors, whereas high occupational stress was associated with more burnout symptoms, sleep complaints and an overall increase in psychological distress. However, there were no effects of CRF on the relationship between occupational stress and the examined health outcomes. Overall, the study appears to be appropriately done and written carefully. However, there are several concerns to this manuscript.

Response: We want to express our thanks for the overall appreciative commentary on our manuscript and the work invested. We regard the comments and suggestions of reviewer #1 as constructive and a useful opportunity to improve the manuscript.

Although there were no modulatory effects of CRF on the relationship between occupational stress and the examined health outcomes, it is unclear why the authors described this statement in their conclusion: “to increase cardiovascular health, including fitness tests in routine health checks and promoting physical activity to further increase cardiorespiratory fitness appears worthwhile.” The conclusion should be modified based on the results from this study.

Response: Thank reviewer #1 for his/her comment. Our statement is based on the fact that we found main effects of cardiorespiratory fitness (CRF) on almost all cardiovascular health risk markers. With this statement, we did not intend to suggest moderating effects of CRF on the relationship between occupational stress and cardiovascular health. Rather, we wanted to suggest that independent of participants’ stress levels, CRF appeared to be an important correlate of cardiovascular health. We have highlighted this more clearly in the revised conclusion section (section 5).

Personally, I think this manuscript would benefit from a mediation analysis.

Response: We agree with reviewer #1 that mediation analyses are interesting and valuable if researchers are interested in testing processes/mechanisms through which one variable impacts on a second variable. However, the focus of our article was on the question whether the relationship between occupational stress on health outcomes depends on participants’ CRF. Such stress-buffering effects are typically examined with moderation (and not mediation) analyses. Moreover, to test “real” mediation effects, longitudinal data on at least three measurement occasions are needed, which unfortunately we do not have.

Furthermore, the authors should consider examining the cross-sectional associations of occupational stress and health outcomes with CRF, and additionally, analysing if CRF modulated these associations. 

Response: We thank reviewer #1 for his/her comment. Nevertheless, we are not sure if we understand this comment correctly. In our study, we examined the cross-sectional associations of occupational stress and CRF with physical and mental health outcomes. As suggested by reviewer #1, our analyses go beyond the analysis of main effect, and test whether CRF modulates the association between occupational stress and participants’ health. The results of these analyses are presented in Tables 2 and 3 (last column; Interaction between stress and CRF).  In the main manuscript, these results are described in section 3.3 (main and interaction effects).

Page 3, lines 46-47, the authors stated: “if participants were not able/allowed to take part in the CRF test, they still could perform all other tests.” It is not clear whether all subjects did all tests. If not, please provide the number of participants on each variable in table 1.

Response: We thank reviewer #1 for his/her comment. As mentioned in section 2.3 (statistical analyses) we have used expectation maximation (EM) algorithm to maximize statistical power. Thus, we had “complete” data for all 201 participants. We have highlighted this more clearly in Table 1, by adding information about the “n”.

How many hours were the participants fasted because fasting blood was drawn? At least three hours? If so, is this shorter duration of fasting long enough to be valid for an accurate analysis of lipid profiles? In addition, the authors should provide the test-retest reproducibility of blood sample using finger prick. How did the authors analyze the blood samples (i.e. HbA1c, lipid profiles)?

Response: We want to thank reviewer #1 for his/her comment. HbA1c reflects plasma glucose concentration over an 8- to 12-week period. HbA1c is a convenient diagnostic indicator for diabetes mellitus, because no fasting is required to measure it. With regard to lipid profiles for cardiovascular risk prediction, Langsted and Nordestgaard (2019) highlighted in their review that before 2009, almost all international societies recommended fasting before measuring lipid profiles in their statements and guidelines. However, they also emphasize that this recommendation was mainly based on the increase seen in triglycerides during a fat tolerance test. Current evidence suggests that individuals eat considerably less fat during a normal day, and that nonfasting triglycerides are superior to fasting in predicting cardiovascular risk. Accordingly, Langsted and Nordestgaard (2019) conclude that “to date there is no sound scientific evidence as to why fasting should be superior to nonfasting when evaluating a lipid profile for cardiovascular risk prediction”.  This is in line with Mora’s (2016) conclusion in a recent JAMA review that “lipid levels differed minimally when measurements were performed nonfasting or fasting, with clinically insignificant changes”. In our study, however, we also assessed body composition via body impedance. For this test, fasting is required to obtain accurate results. Therefore, we asked participants not to eat and drink three hours prior to the data assessment. Three hours of fasting was deemed sufficient, as in a previous study, body fat percentage did not differ after 3 or 12 hours of fasting (Dehghan & Merchant, 2008). Finally, as suggested, we have added more information about how we analysed the blood samples and the test-retest reproducibility of blood sample via finger prick (see section 2.2.2 on cardiovascular risk markers).

Dehghan, M., & Merchant, A. T. (2008). Is bioelectrical impedance accurate for use in large epidemiological studies? Nutrition Journal, 7, doi: 10.1186/1475-2891-1187-1126.

Langsted, A., & Nordestgaard, B. G. (2019). Nonfasting versus fasting lipid profile for cardiovascularrisk prediction. Pathology, 51, 131-141.

Mora, S. (2016). Nonfasting for routine lipid testing. From evidence to practice. JAMA Internal Medicine, 176, 1005-1006.

In the statistical analysis, the authors should provide the correlations of maximal oxygen uptake with all variables; occupational stress variables (JDC, ERI), cardiovascular risk factors and mental health outcomes (burnout symptoms, sleep complaints) in one table. What kind statistic program did the authors use in the present study?

Response: We thank reviewer #1 for his/her comment. Following reviewer#1's recommendation, we have highlighted in section 2.3 (statistical analyses) that we have used SPSS version 25 for all data analyses. We have also provided the correlation matrix between all independent and dependent variables, as part of Table 1. The correlations are briefly described in the text (see section 3.2).

In Tables 2 and 3, why did the authors control for several separate confounders: (a) age and gender as well as (b) age, gender and employment rate…….. (i)? These confounders should be similar controlled for each dependent variable.  

Response: We thank reviewer #1 for his/her comment. In our initial analyses, we only controlled for confounders that were associated with the respective outcome. However, we have followed reviewer#1's recommendation, and have controlled for all covariates across all analyses. While the general pattern of findings remained unchanged, we found some significant interactions between the ERI ratio and CRF for some of the cardiometabolic risk factors.

There are major inconsistencies in how the results were reported in the results section compared with the results presented in the Tables. It is imperative that the description of the results section be consistent with the numerical results in Tables 1-3.

Response: We thank reviewer #1 for his/her comment. We have added a statement on the prevalence of prediabetes and diabetes in the present sample. Otherwise, we were not able to identify any major inconsistencies in how the results were reported in the results section compared with the results presented in Tables 1-3. If reviewer #1 still finds it difficult to follow the description of our results, please let us know more precisely why reviewer #1 feels that the description of the findings in the text and the results in the Tables do not correspond.

In addition, please provide information about SES, HTN, obesity status, etc. that were described the results section into Table 1. Finally, please include information on JDC, ERI score, fitness levels, etc. into Table 1.

Response: We thank reviewer #1 for his/her comment. In the initial version of Table 1, we have only included the descriptive statistics for the outcome variables. In the revision, we have now followed reviewer #1's recommendation, and also integrated the descriptive statistics for the independent variables (JDC, ERI scores, CRF level). To avoid redundancies, the descriptive statistics for the covariates (gender, age, education, relationship status, children at home, caregiving responsibility, employment rate, supervisor status, shift work status, smoking, and regular intake of medication) are presented as part of the main text (section 3.1; sample description).

In line 16 of the results section, the authors stated “see Figure 1,” but there was no Figure 1 in this present paper.

Response: We thank reviewer #1 for his/her comment. His/her observation was correct. We have now integrated a Figure 1, which graphically represents the interaction between the ERI ration and CRF on overall cardiometabolic risk.

Please provide more detailed information about the characteristics of the participants into table 1. 

The authors did not really touch on sex differences at all in the results or discussion. Minimally, the authors should update their results presented in Tables 2 and 3 to be separated by sex, in additional to those for the entire sample. 

Response: We thank reviewer #1 for his/her comment. Sample characteristics are described in detail in the main text (section 3.1). To avoid redundancies, this information was not included in Table 1. However, due to the limited sample size, including gender as an additional moderator was not possible because otherwise the number of participants per cell would have been below the recommended number of approximately 20 (Simmons et al., 2011). Accordingly, we only considered gender as a covariate. Moreover, please note that gender-based norms were used for the categorization of participants in groups with low, moderate and high fitness levels.

Simmons, J. P., Nelson, L. D., & Simonsohn, U. (2011). False-positive psychology: Undisclosing flexibility in data collection and analysis allows presenting anything as significant. Psychological Science, 22, 1359-1366.

Nobody had diabetes? Any subject used medications which may affect mental or physical health, such as anti-depressive drugs, NSAIDS or antioxidant? Another limitation to be noted is the lack of adjustment for socioeconomic status.

Response:We thank reviewer #1 for his/her comment. We did not systematically measure socioeconomic status, however, we were able to control the analyses for several proxies such as education, employment rate, years of employment and supervisor status. Per reviewer#1's request, we have added information on the prevalence of prediabetes and diabetes in the present sample (see section 3.2). We have also added information regarding how many people regularly take medication (see section 3.1).

I don’t know what you mean by “moderation effects”

Response: We thank reviewer #1 for his/her comment. According to the general literature, a moderation effect exists when a third variable significantly influences the relationship between an independent and a dependent variable. The third variable is called the moderator and influences the direct effect between the two other variables. In analysis of variance, the moderation effect can be expressed as an interaction between an independent variable (in our case: occupational stress) and a factor representing the moderator (in our case: fitness level).

Reviewer 2 Report

Police offers as a group of people who fulfil law enforcement duties protect the public by responding to crimes and a variety of other emergencies. However, policy is known to be a high-risk occupation, which requires highly physical and mental demanding. So, police officers are at high risk of contracting physical and mental health issues.  Therefore, protecting police offers’ health also protects the health and safety of the public. This study examined whether cardiorespiratory fitness moderated the association between occupational stress, cardiovascular risk, and mental health. The findings of this study are interesting and meaningful, and contribute to not only the literature but also to public and government’s understanding about how to better protect these people’s health.

In addition, this study was well designed and implemented. The analyses were appropriately conducted and the findings were well interpreted and discussed.  

My only suggestion for this paper to be improved is that police related literature and background should be included in the introduction. Currently, not a single word about police or police officers is mentioned in the introduction. The purpose of this study should be focused on “police officers” not the general population.

Author Response

Basel, 27 June 2019

Dear Prof. Weldon Luo,

We refer to your e-mail dated 17 June 2019 and thank you and the two external referees very much indeed for the opportunity to submit a revision of the aforementioned manuscript for consideration for publication in the International Journal of Environmental Research and Public Health. We found the comments of the reviewers very helpful and constructive in crafting a revision.

As per your request, we are submitting the document electronically. Below, we respond to each point raised by the reviewers. Please note that in the edited manuscript, we have highlighted all changes using a yellow marker. If we can provide any additional information, or make any additional changes, please do not hesitate to let us know.

Sincerely,

René Schilling (on behalf of all co-authors)

Reviewer #2

********************************************************************************************************************

Police offers as a group of people who fulfil law enforcement duties protect the public by responding to crimes and a variety of other emergencies. However, policy is known to be a high-risk occupation, which requires highly physical and mental demanding. So, police officers are at high risk of contracting physical and mental health issues.  Therefore, protecting police offers’ health also protects the health and safety of the public. This study examined whether cardiorespiratory fitness moderated the association between occupational stress, cardiovascular risk, and mental health. The findings of this study are interesting and meaningful, and contribute to not only the literature but also to public and government’s understanding about how to better protect these people’s health.

Response: We would like to thank reviewer #2 for his/her appreciative commentary on our findings. We are delighted to read that the topic was deemed important from a public health perspective.

In addition, this study was well designed and implemented. The analyses were appropriately conducted and the findings were well interpreted and discussed.

Response: Again, we would like to express our thanks for the recognition of our efforts invested in this study, and the favourable opinion on our draft.

My only suggestion for this paper to be improved is that police related literature and background should be included in the introduction. Currently, not a single word about police or police officers is mentioned in the introduction. The purpose of this study should be focused on “police officers” not the general population.

Response: We thank reviewer #2 for this suggestion. We agree with his/her point of view that we could have included more police related literature in the introduction. We have followed reviewer #2's recommendation and have integrated an entire paragraph in the revised manuscript, to provide more background and to focus the article more on our target population.

Round 2

Reviewer 1 Report

The revised manuscript has now been markedly improved relative to its accuracy, readability, and the soundness of conclusions. Therefore, I have no further questions.